# Adaptive Hierarchical Aggregation for Federated Object Detection

## ABSTRACT

In practical object detection scenarios, distributed data and stringent privacy protections significantly limit the feasibility of traditional centralized training methods. Federated learning (FL) emerges as a promising solution to this dilemma. Nonetheless, the issue of data heterogeneity introduces distinct challenges to federated object detection, evident in diminished object perception, classification, and localization abilities. In response, we introduce a task-driven federated learning methodology, dubbed **A**daptive **H**ierarchical **A**ggregation (FedAHA), tailored to overcome these obstacles. Our algorithm unfolds in two strategic phases from shallow-to-deep layers: (1) Structure-aware Aggregation (SAA) aligns feature extractors during the aggregation phase, thus bolstering the global model's object perception capabilities; (2) Convex Semantic Calibration (CSC) leverages convex function theory to average semantic features instead of model parameters, enhancing the global model's classification and localization precision. We demonstrate *experimentally* and *theoretically* the effectiveness of the proposed two modules respectively. Our method consistently outperforming the state-of-the-art methods across multiple valuable application scenarios from 2.26% to 7.61%. Moreover, we build a real FL system using Raspberry Pis to demonstrate that our approach achieves a good trade-off between performance and efficiency.

## CCS CONCEPTS

• **Computing methodologies** → **Distributed algorithms**; **Object detection**.

## KEYWORDS

Federated Learning, Object Detection, Data Heterogeneity

## 1 INTRODUCTION

In recent years, significant success has been achieved in object detection thanks in part to the large scale of labeled dataset, such as the MS COCO [26], ImageNet [5] and SA-1B [20]. However, in practical scenarios, the traditional centralized training strategy cannot be effectively implemented due to the data silos and privacy protection, such as the General Data Protection Regulation (GDPR) [35]. Therefore, it is imperative to explore new training paradigms for object detection tasks.

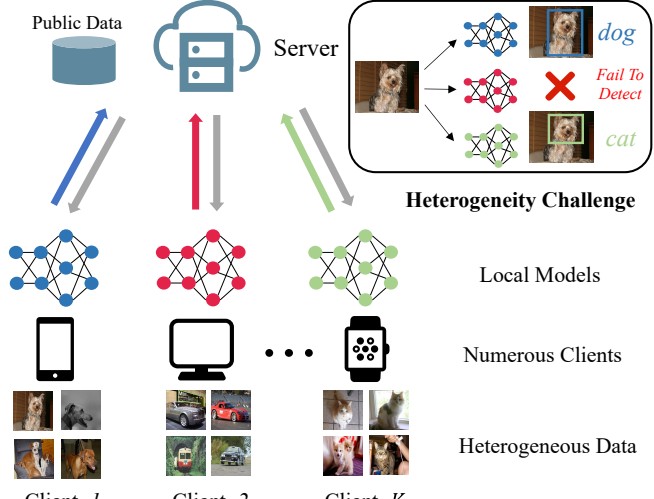

**Figure 1: Illustration of heterogeneous federated learning for object detection in practical scenarios. Data are heterogeneously distributed across numerous clients. Heterogeneity diminishes the model's capacity to perceive objects and ability to classify and locate objects.**

Federated learning (FL), a distributed machine learning paradigm that eliminates the need for data transfer, effectively addresses the limitations mentioned above. It was introduced by McMahan *et al.* along with the FedAvg [33] algorithm and has been extensively explored in the field of computer vision [15, 31, 40, 44, 48]. FL has been an active and challenging research topic and shows promising results in real-world setting [10, 11, 18, 27]. Along with its pilot progress, researches on federated learning are baffled by the key challenge: the issue of data heterogeneity. To address this challenge, some federated learning algorithms based on improvements to FedAvg have been proposed, such as FedProx [24], SCAFFOLD [19] and FedDF [25]. However, these proposed methods have been validated on classification tasks. Unfortunately, federated object detection tasks have received relatively less attention.

Compared to classification tasks, detection tasks face unique challenges when dealing with the issue of data heterogeneity. On one hand, data heterogeneity reduces the global model's ability to perceive objectives. Our experimental analysis has shown that, under task orientation, detection models extract more detailed structural information, which represents the biggest difference from classification models in their patterns of image feature extraction. In FL, heterogeneous data impacts the performance of feature extractors, and further diminishes the global model's capacity to perceive objects. This aspect has been overlooked by many previous researches. On the other hand, the issue of data heterogeneity reduces the model's accuracy in object classification and localization. Beyond

perceiving objects, detection networks must not only classify them but also accomplish localization. However, existing federated algorithms lack consideration for localization accuracy. Thus, achieving satisfactory performance in federated object detection remains a challenge. We further explain these challenges in Figure 1.

To tackle the two aforementioned challenges, we propose our efficient federated learning method tailored for object detection with adaptive hierarchical aggregation: **1) Hierarchical aggregation from shallow-to-deep.** We divide the model into two parts: the feature extractor and the detection head. For each part, we design targeted aggregation strategies, achieving hierarchical aggregation from shallow to deep layers. Our method achieves a thorough integration from texture features to semantic features, effectively addressing the two challenges posed by heterogeneity in federated object detection. **2) Structure-aware Aggregation.** Due to the drift phenomenon [19] between different feature extractors from client models, which leads to a decline in the common model's ability to perceive objects, we introduce a new weighting approach, namely Structure-aware Aggregation, to enhance the consistency of aggregation. Based on image processing and information theory, by combining structural coefficients and informational measures, we design a metric to gauge the consistency among different feature extractors. Through dynamically assigning weights to different client feature extractors based on the consistency measure, our method ensures the stability of the aggregation process, thereby enhancing the global model's ability to perceive objects. **3) Convex Semantic Calibration.** To achieve accurate classification and localization, we need to accomplish high-precision semantic aggregation. Inspired by convex function theory, our focus shifts to averaging the semantic outcomes of the detection head, rather than the parameter averages. Simultaneously, we conduct a detailed analysis of the loss function in object detection, proving that our method is supported by solid and reliable theoretical foundations in mathematics. This strategy not only achieves semantic alignment but also effectively circumvents the problem of client models converging towards diverse local-convex domains, thereby enhancing the performance and robustness of the global model. The main contributions of our paper are as follows:

- Starting from a task-driven perspective, we investigate the unique challenges posed by data heterogeneity in federated learning to object detection tasks. We propose an adaptive hierarchical federated aggregation algorithm that progresses from shallow-to-deep.
- Following rigorous experimental analysis and theoretical validation, we design two modules, Structure-aware Aggregation (SAA) and Convex Semantic Calibration (CSC). SAA enhances the model's object perception capabilities, while CSC improves the model's classification and localization accuracy. Both complement each other effectively.
- We conduct numerous experiments in various datasets, including natural images, remote sensing images and medical images——each representing a valuable federated learning application scenario. Our method achieves superior performance over the state-of-the-art algorithms, and ablation study on core modules validates the efficacy.

- We construct a real federated learning system using Raspberry Pi to authentically verify that our algorithm, by shifting computational overhead to the server side, achieves an excellent trade-off between performance and efficiency.

## 2 RELATED WORK

### 2.1 Object Detection

Object detection has always been a critically important and challenging task in the field of computer vision. In recent years, the rapid development of deep learning techniques has greatly promoted the progress of object detection. Initially, deep learning-based object detection networks were two-stage, such as R-CNN [9] and its variants, Fast R-CNN [8], and Faster R-CNN [36]. Subsequently, end-to-end one-stage networks were proposed. Thanks to their fast detection speed and ease of deployment, they have become the mainstream algorithms in object detection, receiving widespread attention from both the academic and industrial communities. YOLO (You Only Look Once) [34] is a typical representative among them. It was the first one-stage detector in the deep learning era. YOLOv5 [17], as a successor to YOLO, has become one of the most popular object detection networks due to its high accuracy and speed. Recently, YOLOv7 [41], a follow-up work from the YOLOv4 [2] team, has been proposed. It outperforms most existing object detectors in terms of speed and accuracy. In addition, with the growing popularity of Transformer, a Transformer-based object detection network, DETR [3], has been proposed. Transformer discards the traditional convolution operator in favor of attention-alone calculation in order to overcome the limitations of CNNs and obtain a global-scale receptive field. RT-DETR [32], a real-time end-to-end detector based on the DETR architecture, achieves state-of-the-art (SOTA) performance in both speed and accuracy.

### 2.2 Federated Learning

McMahan [33] introduced the concept of federated learning and proposed FedAvg, a pioneering work. The initial design intent of federated learning was to learn information from other data owners without exposing one's own data. In recent years, with increasing attention on privacy concerns, federated learning has gained widespread attention. At the same time, numerous studies have explored the application of federated learning in the field of computer vision, encompassing both CNN and Transformer architectures [1, 16, 22, 28, 45]. He *et al.* [13] proposed a federated learning library and benchmarking framework and evaluated FL on the representative computer vision tasks. It pointed out that in the computer vision domain, model performance in FL is behind centralized training due to the heterogeneity. To mitigate this impact, numerous studies have focused on improving federated learning algorithms. Li *et al.* [24] enhanced the FedAvg algorithm by optimizing the loss function for local client updates and incorporating constraints of the global model on local models. Fang *et al.* [7] optimized the aggregation strategy based on the knowledge distribution of the client models. Zhou *et al.* [47] proposed feature anchors to align the feature mappings and calibrate classifiers across clients during local training. However, all of the above works for the heterogeneity problem are limited to the classification task, and have not been effectively explored on object detection models.

**Figure 2: Illustration of FedAHA, which addresses the heterogeneity issue in federated learning for object detection via shallow-to-deep adaptive aggregation. The Texture Aggregation enhances the global model's perceptual capabilities by dynamically assigning weights to different client models based on mutual information. The Semantic Calibration leverages convex function theory for semantic alignment, enhancing overall coherence.**

Currently, there has been some preliminary exploration in federated object detection [14, 29, 30, 39, 43]. FedVision [29] proposed an engineering platform to support the development of federated learning powered computer vision applications, which is the first real application of FL in computer vision-based tasks. Hegiste *et al.* [14] proposed a FL algorithm for object detection in quality inspection tasks using YOLOv5. However, both of them merely integrate the FedAvg algorithm with object detection tasks, lacking attention to the issue of heterogeneity. Yu *et al.* [43] improved FedAvg based Abnormal Weights Supression, reducing the influence of the weights divergence caused by non-IID and unbalanced data. However, the method of discarding abnormal weights leads to the global model losing some useful information, showing a lack of in-depth solution to the heterogeneity issue. Lu *et al.* [30] and Su *et al.* [39] investigated the issue of cross-domain adaptation in object detection. However, the essence of cross-domain problem still lies in data heterogeneity, and they focused solely on the scenario of autonomous driving, neglecting the algorithm's generalizability.

## 2.3    Discussion

In summary, despite the rich progress made in the field of federated learning within computer vision, there is a lack of effective and in-depth solutions to the issue of heterogeneity in the domain of object detection. Compared to related work, we conduct an in-depth analysis of the specific challenges that heterogeneity poses to object detection, and subsequently, we propose targeted solutions to address these challenges. Additionally, supported by rigorous experiments and theoretical underpinnings, our method exhibits better performance and generalizability.

## 3    METHOD

### 3.1    Problem Setup

Under the standard federated learning setting, there are $K$ clients participating in training, alongside a server that coordinates the aggregation process. Each client trains a local model $\theta_k$ using their private dataset $D_k = \{(x_k^i, y_k^i)\}_{i=1}^{N_k}$, where $N_k$ denotes the size of the $k^{th}$ local dataset, and $(x_k^i, y_k^i)$ represents the inputs and labels, respectively. After every $E$ epochs of local training, all the client models will be transmitted to the central server and aggregated to form a global model. Our objective is to develop a global model with a generalized representation, which can be formulated as follows:

$$\arg\min_{w} L(w) = \sum_{k=1}^{K} \frac{N_k}{N} L_k(w, D_k), \qquad (1)$$

where $N = \sum_{k \in K} N_k$, and $w$ denotes the parameters of the global model, $L_k$ is the loss function of the $k^{th}$ client. In real-world scenarios, private datasets are often heterogeneous, primarily characterized by two factors: an uneven distribution of categories and disparities in data quantities. Given that the server cannot directly access client-side data, to effectively mitigate this issue, similar to FedMD [21], FedDF [25] and RHFL [7], we deploy an unlabeled public dataset $D_0 = \{x^i\}_{i=1}^{N_0}$ on the server side as a bridge for communication between client models. The public dataset facilitates the implementation of our algorithm, and in practical scenarios, it can be easily acquired through publicly available datasets, thereby not introducing additional privacy risks. Moreover, we decompose the detection model parameterized by $w = \{\theta, \phi\}$ into a feature extractor and a detection head. Specifically, the feature extractor extracts shallow texture features of the image and the detection head, given the texture maps, further extracts semantic information and

**input**  **classification**  **detection**

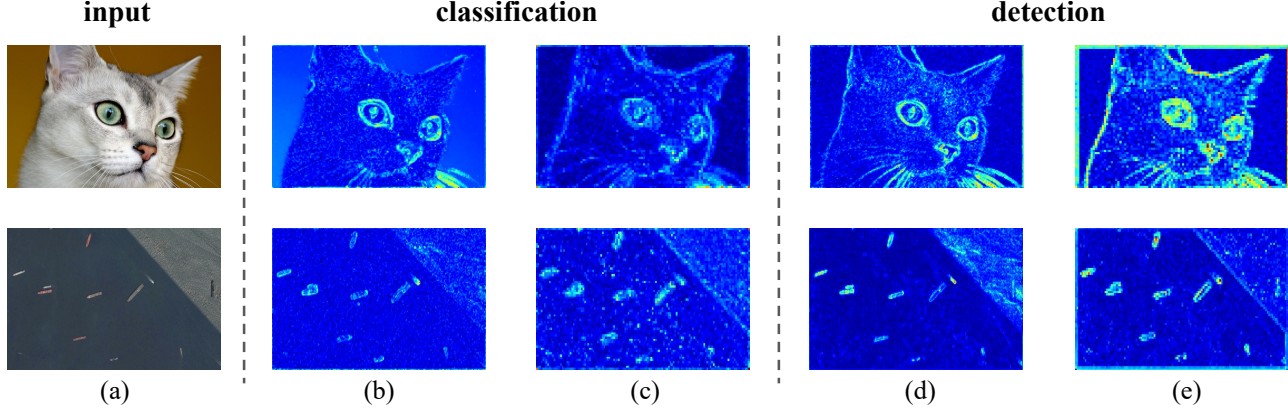

(a)  (b)  (c)  (d)  (e)

**Figure 3: Visualization of the feature maps from the feature extractor for classification and detection tasks: (a) input image, (b) and (c) are output by the classification network, (d) and (e) are output by the detection network. The features are upsampled to the same scale as the input image for comparison. (b) and (d) are low-level features, (c) and (e) are high-level features. As can be seen, the feature maps output by object detection networks contain more image structural information, such as the contours of objects and the edge details of foregrounds and backgrounds.**

predicts results by performing classification and regression. Our work takes into account both the feature extractor and the detection head, achieving a hierarchical federated learning aggregation via shallow-to-deep, as illustrated in Figure 2.

### 3.2 Structure-aware Aggregation

For an in-depth study of the detection task, we visualize the process of the feature extractor extracting texture features and compare it with the classification task, as illustrated in Figure 3. Specifically, for the classification network, we select ResNet50, which boasts a accuracy of 75.99% on the ImageNet dataset. And for the detection network, we use YOLOv5s, which achieves a $mAP_{50}$ precision of 56.8% on the MS COCO dataset. Both networks are thoroughly trained, and the feature extractor employ the same residual network structure. Through the results of visualization, we note that the detection network encompasses more detailed structured image information. This is because detection networks need to detect as fully objects as possible, rather than simply outputting a single result like in classification tasks, which is exactly the unique challenge for the object detection. Unfortunately, in federated learning with data heterogeneity, this challenge is amplified. The feature extractors of heterogeneous client models lead to instability in the aggregation process. To tackle this challenge, taking into account the characteristics of object detection networks, we introduce the Structure-aware Aggregation.

The purpose of the module is to adaptively assign weights to each client model considering their feature extractors, to achieve consistency in model aggregation and mitigate the impacts arising from heterogeneity issues. Inspired by SSIM [42], through combining structural coefficients and informational measures, we implement the measurement of the consistency of feature extractors:

$$M(\theta_k^t) = S(\theta_k^t)I(\theta_k^t). \tag{2}$$

Here, $t$ represents the current round of communication, and $M(\theta_k^t)$ represents the measure of consistency in feature extraction between

the $k^{th}$ client model and the global model, which, in other words, is positively correlated with its corresponding weight.

By comparing the feature maps extracted by client models with those of the global model, we can obtain the structural coefficients $S(\theta_k^t)$ and informational measures $I(\theta_k^t)$, respectively. Suppose $\mathbf{x}$ and $\mathbf{y}$ are two non-negative feature signals from the global and client models, respectively, which have been aligned with each other (e.g., spatial patches extracted from each feature maps). We associate the two normalized vectors $(\mathbf{x} - \mu_x)/\sigma_x$ and $(\mathbf{y} - \mu_y)/\sigma_y$, with the structure of two images, where $\mu$ and $\sigma$ respectively represent the mean and standard deviation. The correlation (inner product) between these is a simple and effective measure to quantify the structural similarity. Notice that the correlation between $(\mathbf{x} - \mu_x)/\sigma_x$ and $(\mathbf{y} - \mu_y)/\sigma_y$ is equivalent to the correlation coefficient between $\mathbf{x}$ and $\mathbf{y}$. Thus we design the structural coefficients as follows:

$$S(\mathbf{x}, \mathbf{y}) = \frac{\sigma_{xy} + \epsilon}{\sigma_x \sigma_y + \epsilon}, \tag{3}$$

where the small constant $\epsilon$ is introduced to avoid division by zero scenarios, specially we set $\epsilon = 0.01$. $\sigma_{xy}$ is the covariance of $\mathbf{x}$ and $\mathbf{y}$, which can be estimated:

$$\sigma_{xy} = \mathbb{E}[(\mathbf{x} - \mu_x)(\mathbf{y} - \mu_y)]. \tag{4}$$

As for information measure, naturally, we think of utilizing mutual information (MI) in information theory, which quantifies the amount of information one variable provides about another. The MI between $\mathbf{x}$ and $\mathbf{y}$ can be calculated as:

$$I(\mathbf{x}, \mathbf{y}) = \mathbb{E}_{p(\mathbf{x}, \mathbf{y})}[log(\frac{p(\mathbf{x}, \mathbf{y})}{p(\mathbf{x})p(\mathbf{y})})], \tag{5}$$

where $p(\mathbf{x})$ and $p(\mathbf{y})$ are the marginal probability distributions of $\mathbf{x}$ and $\mathbf{y}$, and $p(\mathbf{x}, \mathbf{y})$ is their joint probability distribution. By combining the previously obtained structural coefficients and informational measures, we arrive at the final $M(\theta_k)$:

$$M(\theta_k^t) = \mathbb{E}_{(\mathbf{x}, \mathbf{y}_k)}[S(\mathbf{x}, \mathbf{y}_k)I(\mathbf{x}, \mathbf{y}_k)]. \tag{6}$$

Considering the contribution of the number of client datasets to model aggregation, we redefine the aggregation process as follows:

$$\alpha_t^k = \frac{exp(N_k M(\theta_k^t))}{\sum_{i=1}^{K} exp(N_i M(\theta_i^t))}, \tag{7}$$

$$\theta^t = \sum_{k=1}^{K} \alpha_k^t \theta_k^t. \tag{8}$$

### 3.3 Convex Semantic Calibration

To improve the model's precision in classification and localization, we analyze the problem from the perspective of convex optimization theory and introduce a semantic calibration module.

Beyond perceiving objects, the model also needs to process semantic information such as categories and localization. In federated learning, heterogeneous client datasets often lead client models into different local convex domains, which can be formulated as:

$$w_k = \underset{w_k}{\arg\min} \, L_k(w_k; D_k). \tag{9}$$

However, since DNN is indeed a highly non-convex model, the weighted arithmetic average of $K$ local optima does not yield good results, and even the global model might perform even worse than the individual client models. Nevertheless, the loss functions we commonly use, such as cross entropy loss, square loss, and hinge loss, are convex. We will demonstrate the convex properties of the loss function in object detection models in the next section to provide support for our method. Therefore, averaging the semantic information output by client models, rather than their parameters, can guarantee that the global model's performance is at least as good as their average performance, as shown in Figure 4. This is because, when $L(\cdot)$ is a convex function:

$$L(\frac{1}{K} \sum_{k=1}^{K} F(w_k^t; x), y) \leq \frac{1}{K} \sum_{k=1}^{K} L(F(w_k^t; x), y), \tag{10}$$

where $F(\theta; x)$ represents the semantic information output by the model. To this end, we propose to ensemble the client models from the perspective of semantic information. In other words, we aim for the semantic information output by the global model to be as close as possible to the average of the semantic information output by the client models:

$$\phi^t = \underset{\phi^t}{\arg\min} \, ||F(w^t; x), \frac{1}{K} \sum_{k=1}^{K} F(w_k^t; x)||. \tag{11}$$

In order to achieve the effect mentioned above, we design the convex semantic (CS) loss to constrain the semantic information output by the global model as follows:

$$L_{CS}(\phi^t) = \frac{1}{D_0} \sum_{x \in D_0} \frac{1}{K} \sum_{k=1}^{K} ||F(w^t; x), F(w_k^t; x)||_F, \tag{12}$$

where $|| \cdot ||_F$ is Frobenius norm. During the aggregation process, the update of the global model can be denoted as:

$$\phi^t \leftarrow \phi^t - \alpha \nabla_\phi L_{CS}(\phi^t), \tag{13}$$

where $\alpha$ represents the learning rate.

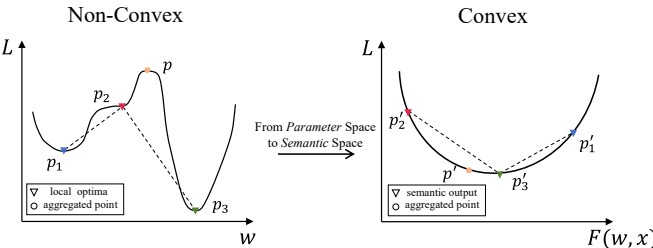

**Figure 4: Illustration of the difference between parameter aggregation and semantic calibration. With the excellent properties of convex functions, semantic calibration enables better performance and generalisation of the model**

### 3.4 Proof of Convex Functions

In this section, we analyze the loss function in object detection to verify convex function property. Specifically, in object detection models, the loss function generally comprises three parts:

$$L = \lambda_{obj} L_{obj} + \lambda_{cls} L_{cls} + \lambda_{loc} L_{loc}. \tag{14}$$

Here, Equation 14 contains confidence loss $L_{obj}$ (judging whether there is an object), classification loss $L_{cls}$, and location loss $L_{loc}$, and the weights $\lambda_{obj}$, $\lambda_{cls}$, and $\lambda_{loc}$ regulate error emphasis among box coordinates, box dimensions, objectness, no-objectness and classification. We will analyze these three parts separately.

**Confidence Loss** assesses the accuracy of the model's prediction regarding the presence of an object within the predicted bounding box. For each predicted box, the model provides a probability prediction of object presence. Since this is a binary classification problem, the commonly used loss function is the cross-entropy:

$$L_{obj}(p, y) = -[y log p + (1 - y) log(1 - p)]. \tag{15}$$

Here, $y$ is the true label, typically 0 or 1, and $p$ is the probability of being predicted as the positive class, ranging between 0 and 1. In the binary cross-entropy function, for some fixed $y$, the second-order partial derivative with respect to $p$ is:

$$\frac{\partial^2 L_{obj}}{\partial p^2} = \frac{y}{p^2} + \frac{(1 - y)}{(1 - p)^2}. \tag{16}$$

This derivative is always positive within the domain of $p$, indicating that the function is convex with respect to $p$.

**Classification loss** assesses the model's ability to correctly classify objects within bounding boxes. This part of the loss focuses on the capability to accurately label detected objects. The multi-class problem is an extension of the binary classification problem, so the loss function here is quite similar to the previously mentioned confidence loss:

$$L_{cls} = \sum_{c=1}^{M} y_c log(p_c), \tag{17}$$

which $M$ represents the number of categories. Therefore, $L_{cls}$ can be expressed as the sum of $M$ convex functions. Assuming $f_1$ and $f_2$ are two convex functions and $f = f_1 + f_2$, $\forall \theta \in (0, 1)$ based on

the properties of convex functions, we can conclude that:

$$
\begin{aligned}
f(\theta x_1 + (1-\theta)x_2) \\
= f_1[\theta x_1 + (1-\theta)x_2] + f_2[\theta x_1 + (1-\theta)x_2] \\
\leq \theta f_1(x_1) + (1-\theta)f_1(x_2) + \theta f_2(x_1) + (1-\theta)f_2(x_2) \\
= \theta f(x_1) + (1-\theta)f(x_2),
\end{aligned}
\tag{18}
$$

which can be easily extended to the sum of $M$ functions. This indicates that $L_{cls}$, a function composed of $M$ convex functions, is also convex.

**Location loss** is used to measure the difference between the predicted bounding boxes and the actual bounding boxes. It includes IoU loss, which measures the degree of overlap between the predicted and actual boxes, and L2 loss, used to minimize the squared error between the coordinates of the predicted and actual boxes. L2 loss is a standard convex function, so we will not expand on its analysis here. Instead, we focus our study on the IoU loss.

In the location process, the output of the object detection model includes four normalized parameters, which we denote as $\tilde{x}, \tilde{y}, \tilde{w}$ and $\tilde{h}$. Here, $\tilde{x}$ and $\tilde{y}$ represent the center coordinates of the predicted bounding box, while $\tilde{w}$ and $\tilde{h}$ represent the width and height, respectively. Inspired by previous research work on IoU loss [37, 46], we decompose the localization problem into two fine-tuning subproblems: translation and scaling.

*For the first sub-problem,* we focus only on the impact of the outputs $\tilde{w}$ and $\tilde{h}$. Considering that client models in federated learning converge to local optima, it is ensured that there will be a significant overlap between the predicted bounding boxes and the ground truth. Therefore, we approximate the IoU as the ratio of the area of the two of them, and the IoU is adjusted only within a small range. Next, we need to analyze the two different expressions of IoU separately. When the predicted bounding box is slightly larger than the ground truth, we simplify the loss function based on this assumption as follows:

$$
L_1(\tilde{w}, \tilde{h}) \approx 1 - \frac{wh}{\tilde{w}\tilde{h}}, \tag{19}
$$

Here, $w$ and $h$ represent the width and height of the bounding box in the ground truth, respectively. Assume $p_1 = (\tilde{w}_1, \tilde{h}_1)$ and $p_2 = (\tilde{w}_2, \tilde{h}_2)$ are two points in its domain of definition, and the latter's box is larger than the former's, i.e., $\tilde{w}_2 > \tilde{w}_1, \tilde{h}_2 > \tilde{h}_1$. Ignoring terms of third order and higher regarding $\theta$ and $(1-\theta)$, we can obtain:

$$
\begin{aligned}
L_1[\theta p_1 + (1-\theta)p_2] - [\theta L_1(p_1) + (1-\theta)L_1(p_2)] \\
= \frac{\theta wh}{\tilde{w}_1 \tilde{h}_1} + \frac{(1-\theta)wh}{\tilde{w}_2 \tilde{h}_2} - \frac{wh}{[\theta \tilde{w}_1 + (1-\theta)\tilde{w}_2][\theta \tilde{h}_1 + (1-\theta)\tilde{h}_2]} \\
\approx 3\theta(\theta-1)\tilde{w}_1 \tilde{h}_1 \tilde{w}_2 \tilde{h}_2 \leq 0.
\end{aligned}
\tag{20}
$$

And when the predicted bounding box is slightly smaller than the ground truth, the loss function can be written as:

$$
L_1(\tilde{w}, \tilde{h}) \approx 1 - \frac{\tilde{w}\tilde{h}}{wh}. \tag{21}
$$

Following Equation20, we can obtain:

$$
\begin{aligned}
L_1[\theta p_1 + (1-\theta)p_2] - [\theta L_1(p_1) + (1-\theta)L_1(p_2)] \\
= \frac{1}{wh}\theta(1-\theta)(\tilde{w}_1 - \tilde{w}_2)(\tilde{h}_2 - \tilde{h}_1) \leq 0.
\end{aligned}
\tag{22}
$$

Up to this point, we have proven the convex property of the loss function in the first sub-problem under two scenarios.

*For the second sub-problem,* we consider the prediction of the bounding box center, focusing solely on the impact of the outputs $\tilde{x}$ and $\tilde{y}$. For convenience of description, we denote the offset between the predicted coordinates $(\tilde{x}, \tilde{y})$ and the ground truth $(x, y)$ as $(\Delta x, \Delta y)$, then we have:

$$
L_2(\tilde{x}, \tilde{y}) = \frac{(h - \Delta y)(w - \Delta x)}{wh + h\Delta x + w\Delta y - \Delta x \Delta y}. \tag{23}
$$

Further, we can calculate its Hessian matrix:

$$
H(L_2) = \begin{bmatrix} \frac{4wh(h-\Delta y)^2}{(wh+h\Delta x+w\Delta y-\Delta x\Delta y)^3} & \frac{2wh(3wh-h\Delta x-w\Delta y+\Delta x\Delta y)}{(wh+h\Delta x+w\Delta y-\Delta x\Delta y)^3} \\ \frac{2wh(3wh-h\Delta x-w\Delta y+\Delta x\Delta y)}{(wh+h\Delta x+w\Delta y-\Delta x\Delta y)^3} & \frac{4wh(w-\Delta x)^2}{(wh+h\Delta x+w\Delta y-\Delta x\Delta y)^3} \end{bmatrix},
\tag{24}
$$

which is a positive-definite matrix. Overall, although the IoU loss is not strictly a convex function, we have demonstrated that it exhibits properties akin to those of a convex function within the context of the problem we considered.

# 4 EXPERIMENTS

## 4.1 Experimental Setting

**Datasets and Models.** We extensively evaluate our method with three public datasets: VOC 2007 [6], NWPU VHR-10 [4], BCCD [38]. These three datasets originate respectively from the realms of natural imagery, remote sensing imagery, and medical imagery, each representing valuable federated learning application scenario. In partitioning heterogeneous datasets, we refer to the commonly used partitioning methods [23] within the federated learning community and combined them with the characteristics and application scenarios specific to each dataset. Specifically, **for the natural imagery VOC 2007,** we adopt the 'label distribution skew' partitioning method. This involves partitioning data for each client based on the labels of the samples. With $K$ clients and $M$ data categories, each client is allocated $M/K$ categories, ensuring no overlap of samples among different clients. **For the remote sensing imagery NWPU VHR-10,** we combine 'feature distribution skew' and 'quantity skew' for data partitioning. Initially, images are categorized and ordered based on the scene of capture (such as airports, industrial areas, residential areas, green zones, oceans, etc.). Subsequently, we introduce the Dirichlet distribution for quantity partitioning, allowing us to flexibly alter the level of imbalance by adjusting the parameter $\beta$. **For the medical imagery BCCD,** we implement the 'quantity skew' partitioning method using the Dirichlet distribution. In addition, we conduct experiments in two levels of heterogeneous scenarios, setting the number of clients $K$, to 4 and 10, respectively, and the parameter $\beta$ in the Dirichlet distribution to 0.5 and 0.1, respectively. Finally, for the models, we select the YOLOv5s and RT-DETER-L, hereinafter abbreviated respectively as YOLOv5 and RT-DETR.

**Evaluation Metrics.** We use mean Average Precision (mAP) to evaluate the detection performance. The mAP is a comprehensive indicator obtained by averaging AP values, which uses an integral method to calculate the area enclosed by the precision-recall curve

**Table 1: Ablation study with *four* heterogeneous clients ($\beta = 0.5$) of different datasets.**

| Model | SAA | CSC | VOC 2007 | | NWPU VHR-10 | | BCCD | |
|---|---|---|---|---|---|---|---|---|
| | | | $mAP_{50}$ | $mAP_{50:95}$ | $mAP_{50}$ | $mAP_{50:95}$ | $mAP_{50}$ | $mAP_{50:95}$ |
| YOLOv5 | | | 72.45 | 46.90 | 91.84 | 61.33 | 85.85 | 58.10 |
| | ✓ | | 73.51 | 47.37 | 91.83 | 61.48 | 86.20 | 58.03 |
| | | ✓ | 72.97 | 47.98 | 91.97 | 61.83 | 85.58 | 58.73 |
| | ✓ | ✓ | **74.72** | **48.79** | **92.77** | **62.97** | **87.82** | **60.25** |
| RT-DETR | | | 73.53 | 48.20 | 90.57 | 60.27 | 84.23 | 57.89 |
| | ✓ | | 74.72 | 49.67 | 90.86 | 61.02 | 85.66 | 58.18 |
| | | ✓ | 73.88 | 48.55 | 90.41 | 61.11 | 84.81 | 59.07 |
| | ✓ | ✓ | **76.34** | **50.12** | **91.49** | **61.51** | **86.50** | **61.81** |

and coordinate axis of all categories. The mAP can be calculated as:

$$mAP = \frac{AP}{M} = \frac{\int_0^1 p(r)dr}{M}, \qquad (25)$$

where $p$ denotes Precision, $r$ denotes Recall, and $M$ is the number of categories. Typically, the mAP is calculated as the average at a specific Intersection over Union (IoU) threshold. Consequently, we employed two widely used thresholding approaches to calculate the mAP: $mAP_{50}$ represents for IoU=0.5 and $mAP_{50:95}$ for increasing IoU threshold values, from 0.5 to 0.95 by 0.05.

**Implementation Details.** Our proposed approach is implemented in PyTorch and runs on a workstation with four NVIDIA 3090 GPUs, using PyTorch-1.8, running Ubuntu 18.04. In local learning process, all clients adopt the same hyper-parameter setting. The input image is resized to 512×512. We use the SGD optimizer with an initial learning rate of $lr$ = 0.01 and the batch size of 256. For each communication round, each client locally trains for 20 epochs. During the local training, we use linear learning rate decay, and after 20 epochs training, the learning rate decays to the original 0.1.

**State-of-the-Art Methods.** In order to prove the effectiveness of our FedAHA, we compare FedAHA with the heterogeneous FL algorithm: FedProx [24], SCAFFOLD [19], FedDF [25], FedAvg(AWS) [43] and FedDAD [30]. Among them, FedProx, SCAFFOLD, and FedDF are classic algorithms targeted at data heterogeneity, while FedAvg(AWS) and FedDAD are federated object detection algorithms designed to mitigate issues of heterogeneity. It is noteworthy that both FedDF and our method incorporate a public dataset within the framework. To ensure a fair comparison, we set aside 10% of the training data from the clients to form a public dataset on the server, thereby guaranteeing that the total amount of data used across all methods is consistent.

### 4.2 Ablation Study

**Effect of Components.** Table 1&2 present the ablation results of our proposed components. Evidently, the SAA and CSC modules each independently contribute to improving the model's performance to a certain extent. A noteworthy observation is that, typically, the use of SAA alone tends to yield a greater enhancement in performance compared to the exclusive use of CSC. This suggests that SAA plays a more fundamental and critical role, aligning with our original design intentions. However, the contribution of

**Table 2: Comparison with the state-of-the-art methods when the number of clients is set to four ($\beta = 0.5$).**

| Model | Method | VOC 2007 | | NWPU VHR-10 | | BCCD | |
|---|---|---|---|---|---|---|---|
| | | $mAP_{50}$ | $mAP_{50:95}$ | $mAP_{50}$ | $mAP_{50:95}$ | $mAP_{50}$ | $mAP_{50:95}$ |
| YOLOv5 | FedAvg | 72.45 | 46.90 | 91.84 | 61.33 | 85.85 | 58.10 |
| | FedProx | 71.36 | 46.68 | 91.29 | 61.42 | 86.06 | 58.87 |
| | SCAFFOLD | 71.48 | 46.51 | 89.30 | 60.59 | 85.49 | 59.02 |
| | FedDF | 72.16 | 47.38 | 90.79 | 61.87 | 86.34 | 58.72 |
| | FedAvg(AWS) | 71.14 | 46.36 | 91.30 | 61.99 | 86.88 | 59.25 |
| | FedDAD | 72.40 | 47.39 | 91.09 | 61.02 | 86.42 | 58.63 |
| | **Ours** | **74.72** | **48.79** | **92.77** | **62.97** | **87.82** | **60.25** |
| RT-DETR | FedAvg | 73.53 | 48.20 | 90.57 | 60.27 | 84.23 | 57.89 |
| | FedProx | 72.53 | 47.63 | 90.25 | 60.32 | 84.71 | 57.45 |
| | SCAFFOLD | 74.74 | 48.32 | 91.04 | 60.25 | 85.73 | 59.54 |
| | FedDF | 74.99 | 49.26 | **91.52** | 61.36 | 84.07 | 59.53 |
| | FedAvg(AWS) | 71.83 | 47.23 | 90.78 | 60.85 | 84.04 | 57.21 |
| | FedDAD | 72.16 | 47.81 | 90.55 | 61.13 | 84.84 | 58.27 |
| | **Ours** | **76.34** | **50.12** | 91.49 | **61.51** | **86.50** | **61.81** |

the CSC module cannot be overlooked. It fine-tunes the model's classification and localization capabilities, which is reflected in the $mAP_{50:95}$ metric. Furthermore, when both modules operate in tandem, they lead to a significant improvement. For instance, when $K = 10$, for the VOC 2007 dataset, the combined action of both modules resulted in a 6.51% increase in $mAP_{50}$, which far exceeds the 3.02%, 1.77% enhancement observed with either module working alone. Thus, the SAA and CSC complement each other effectively.

**Impact of Heterogeneity.** As shown in the Table 1&3, our method can achieve great improvement under different heterogeneity rates, and the improvement will be more obvious at high heterogeneity rates, especially on the VOC 2007 dataset. When the number of clients is four, our method improves on the original baseline by 2.27% $mAP_{50}$ score and 2.81% $mAP_{50}$ score for YOLOv5 and RT-DETR, respectively. When the number of clients increased to ten, our method results in a 6.12% and 6.51% $mAP_{50}$ enhancement over the baseline for YOLOv5s and RT-DETR, respectively. The addition of the SAA component makes the knowledge of relatively noisy clients be learned less in the process of collaborative learning, so our approach demonstrates more significant effectiveness. This demonstrates that the algorithm we proposed is highly effective in mitigating issues related to heterogeneity.

**Table 3: Ablation study with *ten* heterogeneous clients ($\beta = 0.1$) of different dataset.**

| Model | SAA | CSC | VOC 2007 | | NWPU VHR-10 | | BCCD | |
|---|---|---|---|---|---|---|---|---|
| | | | mAP$_{50}$ | mAP$_{50:95}$ | mAP$_{50}$ | mAP$_{50:95}$ | mAP$_{50}$ | mAP$_{50:95}$ |
| YOLOv5 | | | 54.61 | 34.58 | 83.25 | 53.91 | 73.37 | 43.42 |
| | ✓ | | 58.41 | 36.81 | 85.97 | 55.21 | 76.20 | 45.03 |
| | | ✓ | 56.59 | 36.17 | 84.65 | 54.74 | 76.77 | 45.27 |
| | ✓ | ✓ | **60.73** | **39.48** | **87.66** | **56.63** | **78.30** | **46.92** |
| RT-DETR | | | 55.29 | 34.85 | 84.64 | 53.99 | 74.62 | 43.91 |
| | ✓ | | 58.31 | 38.07 | 87.79 | 55.31 | 75.98 | 45.01 |
| | | ✓ | 57.06 | 37.57 | 85.57 | 54.54 | 75.69 | 45.16 |
| | ✓ | ✓ | **61.80** | **38.73** | **87.59** | **56.43** | **77.82** | **46.68** |

**Table 4: Comparison with the state-of-the-art methods when the number of clients is set to ten ($\beta = 0.1$).**

| Model | Method | VOC 2007 | | NWPU VHR-10 | | BCCD | |
|---|---|---|---|---|---|---|---|
| | | mAP$_{50}$ | mAP$_{50:95}$ | mAP$_{50}$ | mAP$_{50:95}$ | mAP$_{50}$ | mAP$_{50:95}$ |
| YOLOv5 | FedAvg | 54.61 | 34.58 | 83.25 | 53.91 | 73.37 | 43.42 |
| | FedProx | 53.12 | 33.95 | 84.36 | 54.63 | 74.35 | 43.82 |
| | SCAFFOLD | 55.17 | 34.76 | 86.28 | **56.71** | 74.94 | 44.98 |
| | FedDF | 56.88 | 36.10 | 85.37 | 54.13 | 76.48 | 45.44 |
| | FedAvg(AWS) | 55.61 | 35.38 | 85.74 | 55.71 | 75.33 | 44.29 |
| | FedDAD | 55.19 | 34.87 | 85.80 | 55.80 | 75.76 | 45.01 |
| | **Ours** | **60.73** | **39.48** | **87.66** | 56.63 | **78.30** | **46.92** |
| RT-DETR | FedAvg | 55.29 | 34.85 | 84.64 | 53.99 | 74.62 | 43.91 |
| | FedProx | 55.10 | 33.91 | 86.12 | 55.71 | 72.27 | 42.69 |
| | SCAFFOLD | 56.86 | 34.04 | 85.48 | 56.02 | 74.72 | 44.71 |
| | FedDF | 57.30 | 35.53 | 85.64 | 55.85 | 75.44 | 43.70 |
| | FedAvg(AWS) | 55.39 | 34.38 | 85.26 | 55.78 | 74.46 | 43.28 |
| | FedDAD | 55.62 | 35.76 | 85.08 | 55.53 | 74.31 | 44.72 |
| | **Ours** | **61.80** | **38.73** | **87.59** | **56.43** | **77.82** | **46.68** |

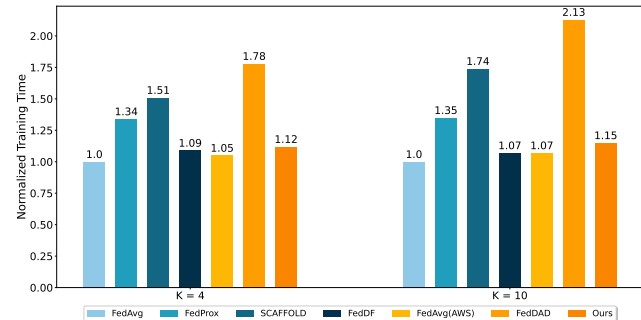

**Figure 5: The average normalized training time of different algorithms in each round of federated learning. This demonstrates that our algorithm achieves an excellent trade-off between performance and efficiency**

## 4.3 Comparison with State-of-the-Art Methods

**Performance Analysis.** We provide comparison results with state-of-the-art methods on two models with three types of data. The comparisons on two heterogeneity rates ($K = 4, 10$) are shown in Table 2&4. Experimental results indicate that our method outperforms existing approaches in most performance metrics under various degrees of heterogeneity. Particularly in scenarios with stronger heterogeneity, our method surpasses other algorithms by a greater margin.

**Robustness Analysis.** Thanks to its task-driven design, our approach has shown both strong model generalization capabilities and dataset generalization abilities. Specifically, our method provides strong support for two mainstream models: the CNN-based YOLOv5 and the Transformer-based RT-DETER. Moreover, our method is capable of handling a variety of datasets, including natural images, remote sensing images, and medical images, which vary significantly in style.

**Efficiency Analysis.** In practical scenarios, the computational capability of clients are typically quite limited. In contrast, servers possess robust computational resource. Therefore, to enhance efficiency in the federated learning process, it is advisable to shift computational overhead to the server as much as possible. Aiming to authentically evaluate the runtime of each algorithm, we utilize the FedML [12] open-source library to establish a small-scale federated learning system composed of Raspberry Pis and

one workstation. To more intuitively compare the efficiency gaps between different algorithms, as shown in Figure 5, we record the training time of various algorithms during each training round and normalize the data. Compared to algorithms like FedProx, SCAFFOLD, and FedDAD, which introduce additional computational or communication overhead on clients, our method exhibits a clear efficiency advantage. For FedDF and FedAvg(AWS), our method can also achieve comparable efficiency performance in the case of significant performance advantages. In summary, our approach achieves a good trade-off between performance and efficiency.

## 5 CONCLUSION

In this paper, we introduce FedAHA, a pioneering federated learning algorithm specifically designed for object detection within heterogeneous scenarios. To address the distinctive challenges encountered in federated object detection, we have meticulously developed two modules: SAA and CSC. Our approach is thoroughly backed by comprehensive experimental analyses and robust theoretical proofs. Extensive testing demonstrates that our method not only outperforms current state-of-the-art methods but also achieves an exceptional balance between performance and efficiency. We are confident that our contributions will significantly broaden the scope of federated learning applications in object detection.

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
