# OpenReview forum: "Adaptive Hierarchical Aggregation for Federated Object Detection"
_acmmm.org/ACMMM/2024/Conference — MM2024 Poster_

### Official Review · Reviewer_9gow · 2024-05-09

**Rating:** 5
**Confidence:** 3

**Summary:**

This paper introduced a task-driven federated learning method called  Adaptive Hierarchical Aggregation (FedAHA) to solve data heterogeneity, non-iid issues for object detection.  To this end, the FedAHA algorithm developed two new techniques namely Structure-aware Aggregation and Convex Semantic Calibration. SAA aligns feature extractors during the aggregation phase, thus bolstering the global model's object perception capabilities. While, CSC leverages convex function theory to average semantic features instead of model parameters, enhancing the global model's classification and localization precision. FedAHA consistently outperforms the state-of-the-art methods across multiple valuable application scenarios from 2.26% to 7.61%.

**Strengths:**

The paper solves an interesting problem of developing an FL algorithm that can work on object detection tasks in non-iid scenarios.  Object detection is often an overlooked task in FL settings.

The paper is well-written and structured.

The proposed framework is implemented on Raspberry Pi devices.

The two techniques named SAA and CSC are novel and are supported by strong theoretical foundations.

The choice of the datasets, models, and baselines seems appropriate.

**Limitations:**

The requirement to have a public dataset is a major limitation.

The performance of FedAHA is not evaluated on the Coco dataset.

It is unclear why evaluation is done on 4 and 10 clients. This also questions the scalability of FedAHA. How will it scale with 50 and 100 clients?

The reason for FedDF outperforming FedAHA on NWPU VHR-10 in Table 2 and SCAFFOLD in Table 4 is not explained.

Section 4.3 on efficient analysis is missing many details. For example, which Raspberry PI model was evaluated? How did you calculate the normalized training time? How are these devices communicating with the server? How much run-time memory do the clients consume?

**Suitability:**

2

---

### Official Review · Reviewer_cBdu · 2024-05-14

**Rating:** 4
**Confidence:** 1

**Summary:**

The paper presents "Adaptive Hierarchical Aggregation" (FedAHA), a novel approach to address challenges in federated object detection caused by data heterogeneity. The approach comprises two main strategies: Structure-aware Aggregation (SAA), which aligns feature extractors to improve object perception, and Convex Semantic Calibration (CSC), which uses convex function theory to enhance classification and localization accuracy. The method involves a hierarchical aggregation process from shallow to deep network layers. The effectiveness of this approach is demonstrated through experiments across different datasets, and the paper includes a practical implementation on Raspberry Pi devices to showcase real-world applicability.

**Strengths:**

1. Novel Methodology: The hierarchical aggregation approach is innovative, addressing specific challenges in federated object detection like feature misalignment and semantic inconsistency due to data heterogeneity.

2. Practical Implementation: The implementation on low-resource devices like Raspberry Pi and the adjustments for computational efficiency make this research highly relevant for real-world applications, particularly in edge computing environments.

3. Theoretical Foundation: The paper provides a solid theoretical foundation for the methodologies used, particularly the application of convex function theory in CSC, which is well-explained and justified.

**Limitations:**

1. Complexity and Resource Demand: While the paper addresses computational efficiency, the hierarchical and dual-phase nature of the method might still impose significant computational demands, particularly on very large federated networks.

2. Generalization to Other Domains: The experiments are limited to object detection; thus, the generalizability of the approach to other domains or more complex tasks within federated learning remains uncertain.

3. Dependence on Accurate Domain Characterization: The success of the method hinges significantly on the accurate characterization of data heterogeneity and appropriate configuration of the aggregation strategies, which might be challenging in highly dynamic environments.

**Suitability:**

2

---

### Official Review · Reviewer_VANg · 2024-05-18

**Rating:** 4
**Confidence:** 3

**Summary:**

The paper introduces a task-driven federated learning methodology,  dubbed Adaptive Hierarchical Aggregation (FedAHA), aimed at the task of target detection in distributed data and strict privacy-preserving scenarios. The algorithm unfolds in two strategic phases from shallow-to-deep layers: (1) Structure-aware Aggregation (SAA) aligns feature extractors during the aggregation phase, thus bolstering the global model’s object perception capabilities; (2) Convex Semantic Calibration (CSC) leverages convex function theory to average semantic features instead of model parameters, enhancing the global model’s classification and localization precision. The proposed method consistently outperforms the state-of-the-art methods across multiple valuable application scenarios.

**Strengths:**

(1)	The proposed algorithm is intuitive and easy to understand and use.

(2)	The proposed algorithm solves the data heterogeneity problem in federated scenarios and enhances the global model’s classification and localization precision.

(3)	The proposed method achieves superior performance over the state-of-the-art algorithms, and ablation study on core modules validates the efficacy.

**Limitations:**

(1)	The authors set aside 10% of the training data from the client to form a public dataset, which has privacy leakage issues.

(2)	The aim of the article is to overcome the existing challenges in terms of object perception, classification, and localization abilities in data heterogeneous environments. mAP is the only metric used in the experiments, which is insufficient to evaluate the object perception, classification, and localization capabilities of the model. I suggest the authors to add indicators that are specific to the evaluation of classification or positioning.

(3)	Lacking comparison with the latest baselines. Among the chosen algorithms, only FedDAD was published in 2023, while the rest of the baselines were proposed in 2020 or earlier.

**Suitability:**

2

---

### Official Review · Reviewer_vymH · 2024-05-24

**Rating:** 4
**Confidence:** 2

**Summary:**

The paper presents a federated learning methodology aimed at addressing the challenges posed by data heterogeneity in federated object detection tasks. The proposed approach, termed Adaptive Hierarchical Aggregation (FedAHA), involves two main phases: Structure-aware Aggregation (SAA) and Convex Semantic Calibration (CSC). The SAA phase aligns feature extractors to enhance the global model's object perception capabilities, while the CSC phase leverages convex function theory to average semantic features, improving the global model's classification and localization precision. The authors demonstrate the effectiveness of these modules through theoretical validation and extensive experiments.

**Strengths:**

Clarity: The paper is well-structured and clearly explains the motivation, methodology, and results. The inclusion of detailed visualizations and ablation studies enhances the comprehensibility of the proposed approach.
Theoretical and Technical Correctness: The use of convex function theory in the CSC module is well-grounded and provides a solid mathematical basis for improving semantic feature aggregation. The theoretical validation of the proposed methods adds robustness to the claims.

**Limitations:**

Lack of Novelty in Baseline Comparison: While the proposed method shows improvements, the baselines used for comparison (e.g., FedAvg, FedProx) are relatively standard. It would strengthen the paper to include comparisons with more recent and advanced federated learning techniques specifically designed for object detection.
Limited Analysis of Computational Efficiency: Although the paper mentions efficiency and even implemented a real federated learning system using Raspberry Pis, a more detailed analysis of the computational cost and scalability of the proposed method compared to other state-of-the-art methods would be valuable. This includes the impact on training time and resource utilization in real-world federated learning deployments.
Logical Flow in Introduction: The introduction section mentions the two challenges of applying federated learning (FL) to object detection, but the logic is unclear. It is recommended that the authors first introduce the key factors affecting the performance of object detection before introducing FL. The impact of data heterogeneity in FL on object detection performance is not clearly articulated.

**Suitability:**

2

---

### Meta-Review · Area_Chair_RN9U · 2024-06-26

**Recommendation:** Accept (Poster)
**Confidence:** 5

**Metareview:**

3 BA and 1 WA, this paper can be accepted as poster.